# Bootstrapping quantum process tomography via a perturbative ansatz

L.C.G. Govia [1✉], G.J. Ribeill[1], D. Ristè [1], M. Ware[1] & H. Krovi [1]

Quantum process tomography has become increasingly critical as the need grows for robust verification and validation of candidate quantum processors, since it plays a key role in both performance assessment and debugging. However, as these processors grow in size, standard process tomography becomes an almost impossible task. Here, we present an approach for efficient quantum process tomography that uses a physically motivated ansatz for an unknown quantum process. Our ansatz bootstraps to an effective description for an unknown process on a multi-qubit processor from pairwise two-qubit tomographic data. Further, our approach can inherit insensitivity to system preparation and measurement error from the two-qubit tomography scheme. We benchmark our approach using numerical simulation of noisy three-qubit gates, and show that it produces highly accurate characterizations of quantum processes. Further, we demonstrate our approach experimentally on a superconducting quantum processor, building three-qubit gate reconstructions from two-qubit tomographic data.

[1] Raytheon BBN Technologies, 10 Moulton Street, Cambridge, MA 02138, USA. ✉email: luke.c.govia@raytheon.com

Recent years have seen remarkable progress in quantum information processing, with rapid advancement towards high-fidelity multi-qubit systems[1–3], some of which are now publicly available[4,5]. This has enabled significant achievements in many aspects of quantum computation, such as first demonstrations of the building blocks for error correction and fault-tolerance, e.g., refs. [6–12]. Concurrently, demonstrations of noisy-intermediate-scale quantum algorithms[13] that do not require full fault-tolerance, e.g., refs. [14–18], make real world applications of quantum information processing a near-term possibility.

In light of these achievements, the need for robust, accurate and efficient validation, and verification of quantum processors becomes ever more pressing. This is the natural domain of quantum state tomography (QST) and quantum process tomography (QPT). Respectively, QST and QPT seek to characterize the state of a quantum processor or the dynamical map of its evolution[19]. Unfortunately, naive implementations of both QST and QPT require an experimental effort that scales exponentially with the number of qubits. For practical purposes, this scaling has limited full QST and QPT to small system sizes, e.g., refs. [20,21], though this can be improved using approximate characterizations[22,23], or in situations with large amounts of symmetry[24,25].

Further compounding QPT, the most error-prone operations are often system preparation and measurement (SPAM), which can overwhelm the intrinsic error in high-fidelity quantum processes and hinder their characterization. Several SPAM-insensitive metrics exist, such as the widely successful randomized benchmarking[26–29] and its variants[30–36], as well as gate set tomography (GST)[37–39]. Randomized benchmarking has the additional benefit of overcoming the exponential scaling of standard QPT, but at the cost of returning only a single number characterizing the quantum process.

In this work, we present an approach to efficient QPT that reduces the exponential scaling to quadratic scaling, while still returning a full process matrix describing the quantum process. We propose the Pairwise Perturbative Ansatz (PAPA), which describes the unknown quantum process as sequential two-qubit processes on all qubit pairs. We show how to fit the free parameters of our ansatz to data obtained from QPT of two-qubit subsets of the full system. When this data is provided by SPAM-insensitive tomography, such as GST, our approach becomes SPAM-insensitive as well as efficient.

## Results

### Ansatz for process tomography

A generic $N$-qubit quantum process, which we label as $\mathcal{E}$, has $16^N - 4^N$ free parameters, and determining this exponentially scaling number of free parameters is what makes naive QPT an exponentially hard problem. We propose to restrict the unknown process a priori by assuming an ansatz for its form, which in turn restricts the number of free parameters in the unknown process, allowing for efficient QPT.

Process tomography can be rephrased as state tomography of the Choi dual-state (via the Choi-Jamiołkowski isomorphism), which is the state formed when the unknown process acts on one half of a maximally entangled state in a Hilbert space of dimension $2^{2N}$, given by

$$\rho_{\mathcal{E}} = \frac{1}{2^N} \sum_{\mu\nu} |\psi_\mu\rangle\langle\psi_\nu| \otimes \mathcal{E}\left(|\psi_\mu\rangle\langle\psi_\nu|\right), \tag{1}$$

where $\{|\psi_\mu\rangle\}$ is an orthonormal basis for $N$-qubit Hilbert space.

Thus, one can use efficient state tomography methods for process tomography, such as compressed sensing[22,40,41] and matrix-product-state (MPS) parameterizations[23,42–44]. Unfortunately, the matrix completion algorithms that underlie these approaches can themselves be inefficient in run-time. This issue can be circumvented using constrained approaches, as in refs. [23,43], which restrict to pure state descriptions of the unknown quantum state.

Both compressed sensing and MPS parameterizations implicitly assume an ansatz for the unknown quantum process, that it is either low rank, or has a matrix product structure (and thus short-range correlations), respectively. Our pairwise perturbative ansatz assumes a different physical constraint on the unknown process: that it is intrinsically built from two-qubit processes on all pairs of qubits. Like the MPS approach, this implies that few-body QPT is sufficient to find a PAPA characterization of the unknown process. Unlike an MPS, PAPA has no locality constraint on correlations, and allows for long-range correlations. Further, we will see that the PAPA constraint is physically motivated, unlike the low rank restriction of compressed sensing.

### Pairwise perturbative ansatz

We will assume an ansatz where the unknown $N$-qubit process is written as a composition of two-qubit processes, consisting of quantum processes for each qubit pair in the system. This is most easily expressed in terms of the super-operator matrix representation $\hat{\mathcal{E}}$ of the quantum process $\mathcal{E}$, as the series composition becomes a product of matrices. This has the general form

$$\hat{\mathcal{E}} = \prod_{k=1,n=1}^{N-1,N-k} \hat{\mathcal{E}}_{k,n+k}, \tag{2}$$

where $\mathcal{E}_{k,n+k}$ is an arbitrary two-qubit process on qubits $k$ and $(n+k)$ with no restrictions.

The product runs over all pairs of qubits, of which there are $(N^2 - N)/2$. Each of the unknown two-qubit processes can be written as

$$\mathcal{E}_{k,n+k} = \sum_{i_{k,n}, j_{k,n}}^{16} \chi_{i_{k,n}}^{j_{k,n}} (\mathcal{I}^{\otimes k-1} \otimes \mathcal{A}_{i_{k,n}}^{(k)} \otimes \mathcal{I}^{\otimes n-1} \\ \otimes \mathcal{A}_{j_{k,n}}^{(k+n)} \otimes \mathcal{I}^{\otimes N-k-n}), \tag{3}$$

where $\{\mathcal{A}_{i_{k,n}}^{(k)}\}$ is a complete basis for single-qubit processes and $\mathcal{I}$ is the identity process. $\chi_{i_{k,n}}^{j_{k,n}}$ is an element of the $\chi$-matrix describing the two-qubit process, and the summation variables $i_{k,n}$ and $j_{k,n}$ are subscripted to emphasize that they correspond to a particular qubit pair.

There are many possible ansatze for an unknown quantum process[22,23,40–44], but the form we have chosen is particularly well motivated physically. As it is the composition of two-qubit processes in sequence, it captures the natural two-body quantum operations that occur in a gate-based quantum computation. It can completely specify any ideal gate operation (single-layer quantum circuit built from one and two-qubit gates), and will contain both single-qubit errors and correlated two-qubit errors as independent free parameters. It also describes processes that involve more than two qubits, but as combinations of two-qubit processes performed in sequence. Thus, it describes general processes in a perturbative fashion, built from one- and two-qubit processes.

While each arbitrary two-qubit process described by Eq. (3) is parameterized in terms of a basis with $16^2$ elements, its $\chi$-matrix has only $16^2 - 4^2 = 240$ free parameters. There are $\binom{N}{2} = (N^2 - N)/2$ two-qubit subsets, and so the total number of free parameters in our

ansatz is $120(N^2 - N)$. As this scales quadratically with qubit number, PAPA is an efficient approach to QPT. The order of the pairwise processes in PAPA can be chosen arbitrarily, but the best results will be found using the order of two-qubit gates in the shallow-depth circuit being characterized.

QPT with PAPA consists of determining the $\chi$-matrix for each two-qubit process in the product in Eq. (2). Inspired by the local tomography in refs. [23,43], we use the tomographic characterization of two-qubit processes on all pairs of qubits to determine these free parameters. In essence, from characterization of two-body processes, we bootstrap to a multi-qubit process of PAPA form.

To compare the PAPA ansatz to two-qubit tomographic data, we must determine a notion of a two-qubit reduction of a process $\mathcal{E}$. This is most easily done in terms of the Choi state $\rho_{\mathcal{E}}$. For the two-qubit subset $\mathcal{S} = \{m, p\}$ this takes the form

$$\rho_{\mathcal{S}} = \frac{1}{2^N} \sum_{\mu\nu} \text{Tr}_{/\mathcal{S}} \Big[ |\psi_\mu\rangle\langle\psi_\nu| \Big] \otimes \text{Tr}_{/\mathcal{S}} \Big[ \mathcal{E}\big( |\psi_\mu\rangle\langle\psi_\nu| \big) \Big], \quad (4)$$

where by $\text{Tr}_{/\mathcal{S}}[\rho]$ we mean the partial trace of all qubits other than those in the set $\mathcal{S}$, and it is important to note that the partial trace is applied to both "parts" of the Choi state. Using the orthogonality of the $N$-qubit basis, we see that

$$\text{Tr}_{/\mathcal{S}} \Big[ |\psi_\mu\rangle\langle\psi_\nu| \Big] = \delta_{\mu_{/\mathcal{S}}, \nu_{/\mathcal{S}}} |\psi_{\mu_{\mathcal{S}}}\rangle\langle\psi_{\nu_{\mathcal{S}}}|, \quad (5)$$

where the indices $\mu_{\mathcal{S}}$ ($\mu_{/\mathcal{S}}$) are the subset of indices in $\mu$ that correspond to the qubits inside (outside) of the subset $\mathcal{S}$. Thus, the reduced Choi state of the unknown process can be written as

$$\rho_{\mathcal{S}} = \frac{1}{2^2} \sum_{\mu_{\mathcal{S}}\nu_{\mathcal{S}}} \Bigg( |\psi_{\mu_{\mathcal{S}}}\rangle\langle\psi_{\nu_{\mathcal{S}}}| \otimes \text{Tr}_{/\mathcal{S}} \Big[ \mathcal{E}\Big( |\psi_{\mu_{\mathcal{S}}}\rangle\langle\psi_{\nu_{\mathcal{S}}}| \otimes \frac{\mathbb{I}_{N-2}}{2^{N-2}} \Big) \Big] \Bigg), \quad (6)$$

where $\mathbb{I}_{N-2}$ is the identity matrix of dimension $2^{N-2}$.

To determine the free parameters in the PAPA ansatz, for each pair of qubits we compare $\rho_{\mathcal{S}}$, the two-qubit reduced Choi states described by Eq. (6) with PAPA free parameters, to the corresponding experimentally characterized two-qubit Choi state, which we label $\sigma_{\mathcal{S}}$. This experimentally characterized state is the result of two-qubit QPT performed on the qubit pair $\mathcal{S}$ for the application of the global unknown process $\mathcal{E}$, as depicted in Fig. 1a). We equate this to our reduced Choi state for the unknown process, $\rho_{\mathcal{S}}$, to determine the free parameters in the PAPA. In other words, we simultaneously solve the equations

$$\rho_{\mathcal{S}} = \sigma_{\mathcal{S}}, \quad (7)$$

for every pair of qubits.

To do this, we perform a non-linear least-squares minimization of the cost function

$$C_1\big[ \vec{\chi} \big] = \sum_{\mathcal{S}} \sum_{k,n} \Big| [\rho_{\mathcal{S}}(\vec{\chi})]_{k,n} - [\sigma_{\mathcal{S}}]_{k,n} \Big|^2, \quad (8)$$

as a function of the free parameters of PAPA, i.e., the $\chi$-matrix elements of Eq. (3). This is the element-wise difference between the experimentally characterized two-qubit Choi state $\sigma_{\mathcal{S}}$, and the PAPA reconstruction two-qubit reduced Choi state $\rho_{\mathcal{S}}$, summed over all qubit pairs. Calculation of $\rho_{\mathcal{S}}$ becomes inefficient at large $N$ for Schrödinger-style evaluation of Eq. (4). In future work, we hope to improve performance via parallelization of this calculation using Feynman path integral approaches[45,46]. Further details and pseudocode can be found in Supplementary Notes 2 and 6.

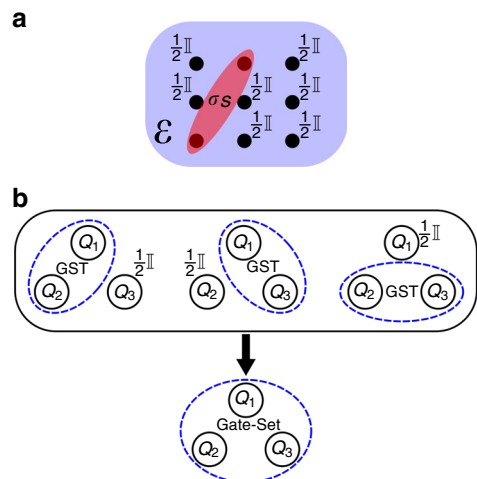

**Fig. 1 Schematic of the pairwise perturbative ansatz. a** Pairwise perturbative ansatz (PAPA) tomography: for all qubit pairs, characterize the effective two-qubit process (Choi state $\sigma_{\mathcal{S}}$) when the unknown $N$-qubit process $\mathcal{E}$ occurs, and all other qubits start in the maximally mixed state. **b** Three-qubit PAPA+GST: characterized two-qubit gate sets are bootstrapped to a three-qubit gate set via PAPA.

The total experimental requirement for PAPA is two-qubit QPT on the $(N^2 - N)/2$ pairs of qubits. Each of the pairwise characterized two-qubit processes is described by $16^2 - 4^2 = 240$ complex numbers, which gives a total of $120(N^2 - N)$ total complex numbers describing the characterization of all qubit pairs. Thus, we have exactly as many constraints (coming from experimental characterization) as there are free parameters in PAPA. This further motivates our choice of ansatz, as we have made use of all available data from two-qubit characterizations of the unknown multi-qubit process.

Note that each $\rho_{\mathcal{S}}$ depends on the $\chi$-matrix elements for all qubit pairs, i.e., those in all $\mathcal{E}_{k,n+k}$, not just the qubit pair of the subset $\mathcal{S}$. Thus, each two-qubit process characterization $\sigma_{\mathcal{S}}$ constrains the global process, not just the component of the ansatz on the qubits in $\mathcal{S}$.

**PAPA and gate set tomography.** The PAPA tomography approach described so far works well to obtain a bootstrapped description of an $N$-qubit process from characterization of the effective processes on all qubit pairs. However, often the problem at hand is not to characterize a completely unknown process, but to determine the actual process, $\mathcal{G}$, that occurs when we aim to implement a unitary gate, $\hat{G}$, (from here on we use calligraphic text for processes and latin text for unitary gates).

Extending this to an entire gate set via gate set tomography (GST), we obtain a set of processes $\{\mathcal{G}_i\}$ corresponding to the experimental implementation of an ideal gate set $\{\hat{G}_i\}$. GST has the further benefit of excluding state-preparation and measurement (SPAM) errors from the processes $\{\mathcal{G}_i\}$[38]. Note that for clarity we will use "gate set" to refer to the processes $\{\mathcal{G}_i\}$, and "ideal gate set" to refer to the unitary gates $\{\hat{G}_i\}$.

Combining PAPA with GST, we can perform GST on all qubit pairs to obtain a characterized gate set for each pair, and then use PAPA to bootstrap to descriptions of $N$-qubit processes. To see why this is useful, consider the three-qubit gate $\hat{X} \otimes \hat{Y} \otimes \hat{X}$. Given characterized gate sets with the relevant two-qubit gates,

one way to describe the three-qubit process would be

$$\hat{X} \otimes \hat{Y} \otimes \hat{X} \rho \hat{X} \otimes \hat{Y} \otimes \hat{X} \rightarrow \mathcal{G}_{X_1 Y_2}\left(\mathcal{G}_{I_2 X_3}(\rho)\right) \qquad (9)$$

where $\mathcal{G}_{AB}$ is the experimental process when we try to implement the gate $\hat{A} \otimes \hat{B}$. However, there is ambiguity in the correct decomposition of the three-qubit gate, and $\mathcal{G}_{X_1 X_3}(\mathcal{G}_{Y_2 I_3}(\rho))$ would be an equally valid description of the process. An issue arises as it is unlikely that the constructed three-qubit processes from all possible two-qubit decompositions will agree with one another.

Using PAPA avoids this issue, as it finds the three-qubit process of PAPA form that best agrees with the pairwise characterized processes, i.e., with $\mathcal{G}_{X_1 Y_2}$, $\mathcal{G}_{Y_2 X_3}$, and $\mathcal{G}_{X_1 X_3}$. As such it captures context dependence between gate operations, such as when the effect on qubit 1 is different for the processes $\mathcal{G}_{X_1 Y_2}$ and $\mathcal{G}_{X_1 X_3}$. As an added benefit, one never has to implement the full $N$-qubit process, as one does when using PAPA without GST (as described in the previous section). Instead, from the characterized gate sets on all qubit pairs, we can bootstrap to PAPA characterizations of the processes in an $N$-qubit gate set (as represented in Fig. 1b).

While PAPA can return a characterization of any $N$-qubit gate, when we restrict the pairwise two-qubit QPT to GST, the PAPA+GST combination can only characterize a limited set of $N$-qubit gates. Which $N$-qubit gates can be characterized with PAPA+GST is gate set dependent, and detailed further in Supplementary Note 3. The general requirement is that each two-qubit reduced process of the ideal $N$-qubit gate must be an incoherent mixture of two-qubit gates built from the ideal gate set. For example, if the ideal gate is a controlled-not gate on qubit pair 1-2, $\mathrm{CNOT}_{12} \otimes \hat{\mathbb{I}}$, then the ideal gates $\hat{Z} \otimes \hat{\mathbb{I}}$ and $\hat{\mathbb{I}} \otimes \hat{\mathbb{I}}$ need to be in the characterized gate set for qubit pair 1–3.

Decomposing an $N$-qubit gate this way implicitly assumes the errors that make the implemented process $\mathcal{G}$ distinct from the ideal gate $\hat{G}$ are not strongly specific to the implementation of $\mathcal{G}$. This is easily satisfied if the errors are gate-independent, but some kinds of gate-dependent error are tolerable, such as context dependence in simultaneous single-qubit gates. For the $\mathrm{CNOT}_{12} \otimes \hat{\mathbb{I}}$ gate considered previously, an example of a tolerable gate-dependent error would be a coherent error that occurs on qubit 1 both for an actual $\hat{Z}$-gate or an effective $\hat{Z}$-gate (as occurs in the reduced process on qubit pair 13 for the $\mathrm{CNOT}_{12}$ gate).

It is important to emphasize that neither of these issues are limitations of PAPA, which can characterize any $N$-qubit process using pairwise two-qubit QPT, but of the two-qubit characterizations supplied to PAPA by GST. Nevertheless, there are many situations where PAPA+GST may be applicable, i.e., the ideal-gate decomposition is possible and the errors can be assumed to be captured by PAPA+GST. In the following sections we explore such a situation in both experiment and theory. For situations where PAPA+GST is not possible, PAPA can inherit SPAM-insensitivity from other SPAM-insensitive process tomography such as that using randomized benchmarking[47–49].

A further subtlety of using PAPA with GST is the intrinsic gauge freedom[38] in GST-derived gate sets. This gauge freedom arises from characterizing the gates, preparation, and measurements simultaneously, and results in a continuous family of gate sets that are consistent with experimental data (see Methods for more details). This becomes an issue in PAPA when the pairwise gate sets are not in the "same" gauge. For instance, the same gate

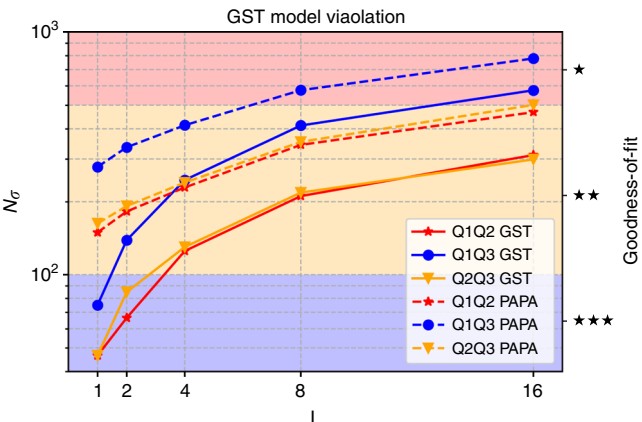

**Fig. 2 GST (solid lines) and PAPA reconstruction (dashed-lines) $N_\sigma$ vs. germ power $L$ for each of the three experimental data sets.** $N_\sigma$ quantifies the deviation from a Markovian qubit model. The goodness-of-fit parameter is provided by pyGSTi and ranges from single blackstar to five black stars, indicating how well the data fits the implicit model. Larger values of $L$ correspond to increased sensitivity to gate error and to longer circuits.

on one qubit may have different descriptions in different pairwise gate sets, with the descriptions related by a gauge transformation.

Many relevant quantities derived from the process matrix description of gates (such as the trace distance used later) are gauge variant. To minimize the error introduced by gauge freedom, GST can perform a gauge optimization of the characterized gate set to the target (ideal) gate set. We find that this is sufficient to ensure that the pairwise gate sets used for PAPA are approximately gauge consistent. Our attempts to further improve the results by additional gauge optimization are detailed in the Methods and Supplementary Note 4.

**Experimental reconstructions**. We demonstrate the PAPA+GST approach experimentally using an IBM five-qubit device similar to that of ref. [9]. For this demonstration, we focus on a three-qubit subset of the chip with the goal of reconstructing three-body operations. Device parameters, and coherence times can be found in Supplementary Note 1. To begin, as described in the Methods, two-qubit GST is performed on all three pairs of qubits in the subset, as is depicted in Fig. 1b. We choose the gate set $\{\hat{X}_{90}, \hat{Y}_{90}, \hat{\mathbb{I}}\}^{\otimes 2}$ composed of simultaneous 90° rotations around the $X$ and $Y$ axes, with the idle gate on both qubits (all 80 ns long). This set is chosen to allow the bootstrapping of non-trivial three-body operations and to avoid the issues discussed in the previous section. We use the GST software package pyGSTi[50] and its std2Q_XXYYII gate set to build experimental gate strings, and to numerically reconstruct the two-qubit gate set characterizations.

Figure 2 shows the $N_\sigma$ standard deviations from GST's implicit qubit model as a function of germ power $L$ for the three data sets (solid lines). The goodness-of-fit parameter on the right axes is supplied by pyGSTi as a rough gauge for how well the model captures the dynamics in the data. It is clear from the figure there are significant deviations at higher germ powers. We attribute this to a combination of drift in system parameters, and leakage into higher excited states.

The final output of the GST algorithm yields three distinct gauge-optimized gate set characterizations, which are used by PAPA to reconstruct the larger three-qubit processes. Each of the 27 three-qubit gates in the gate set are reconstructed using a

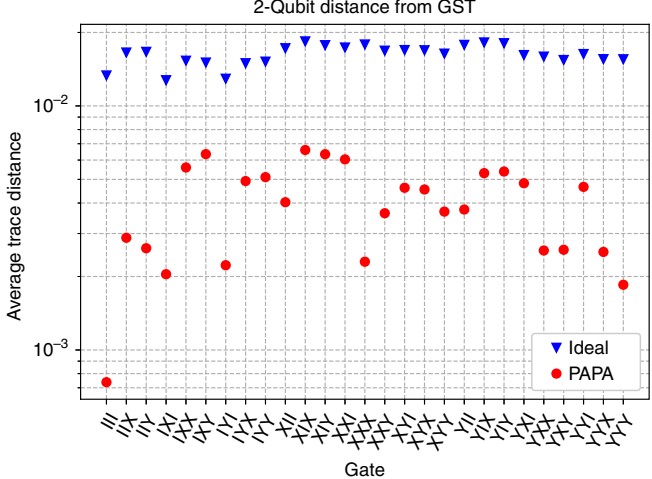

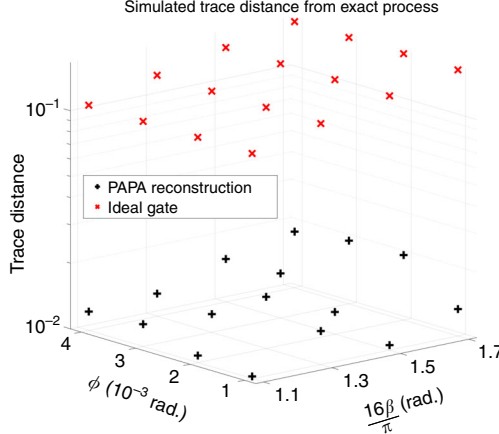

**Fig. 3 Comparison of the GST measured process matrices to the ideal and the PAPA reconstructions of the experimental data.** The PAPA data points are the average trace distance of the three reduced processes from their corresponding GST characterizations. The ideal points are the average trace distance of the ideal gates to GST. The trace distances (Eq. (10)) for the PAPA reconstructions are lower for all 27 gates in the set.

**Fig. 4 Cross resonance gate simulations.** Simulated trace distance between the three-qubit Choi state for the simulated CR-CNOT with coherent error and either the PAPA reconstructed Choi state (black +) or the ideal gate (red ×), as a function of over-rotation error (angle $\beta$) and stray $ZZ$-coupling (angle $\phi$), see Eq. (17).

Julia[51] implementation of the PAPA algorithm. Details of the non-linear least-squares bootstrapping can be found in Supplementary Note 6.

The main experimental result is plotted in Fig. 3. For each three-qubit gate in the reconstructed gate set $\{\hat{X}_{90}, \hat{Y}_{90}, \hat{\mathbb{I}}\}^{\otimes 3}$, we compare the GST characterizations of the effective two-qubit gates on each pair of qubits ($\sigma_{\mathcal{S}}$), to either the ideal reduced two-qubit gate, or the reduced two-qubit process obtained from the PAPA three-qubit reconstruction ($\rho_{\mathcal{S}}$). We quantify the distance between processes using the trace distance

$$\text{Trace Dist.} = \frac{1}{2} \text{Tr}\left[ \sqrt{(\rho_{\mathcal{S}} - \sigma_{\mathcal{S}})^{\dagger}(\rho_{\mathcal{S}} - \sigma_{\mathcal{S}})} \right], \qquad (10)$$

and the average of the three trace distances (one for each reduced two-qubit process) is what is plotted in Fig. 3.

In all cases, PAPA produces an estimate of the process closer to the experimental data (GST characterizations) than the ideal gate. Thus, from two-qubit tomography our PAPA+GST bootstrapping technique has produced a characterization of the three-qubit gate set that is both consistent with the tomography data (small trace distance in Fig. 3), and consistent across pairs of qubits (by nature of the ansatz).

As another comparison to experimental data, we use the model violation metrics from pyGSTi to directly compare how well reduced two-qubit gate sets obtained from the PAPA reconstructed three-qubit gates fit the experimental data. We create a PAPA reduced two-qubit gate set by tracing out the third qubit when the intended gate was identity. As shown in Fig. 2, while the PAPA reconstructions (dashed-lines) have increased model violation, the increase is not extreme.

One may ask if PAPA is producing a characterization that could be explained with a simple model, such as single-qubit decoherence. However, a search over all possible values of $T_1$ and $T_2$ found no values that would make the data consistent with a simple decoherence model, and in fact such models did worse than the ideal gate.

**Simulation tests of the ansatz.** We test the PAPA approach in simulation by examining coherent error in a cross-resonance

(CR) implementation of a CNOT gate[52–54], with the ideal gate taking the form $\text{CNOT}_{12} \otimes \hat{\mathbb{I}}$. Referred to as a CR-CNOT, this ideal gate consists of the ideal CR-gate followed by single-qubit gates. We consider noisy implementations where the CR-gate has coherent error due to unwanted coherent interactions (see Methods).

The error model we consider for CR-CNOT is strongly gate-dependent, since it is intrinsic to the CR interaction itself. However, the reduced two-qubit gate decomposition of the CNOT (see Supplementary Note 3) used in PAPA+GST contains gates that do not involve the CR interaction. These gates will be insensitive to the CR error, and as a result PAPA+GST is not applicable in this situation. Instead we apply standard PAPA, and simulate QPT on the effective process for each pair of qubits during the implemented CR-CNOT. For this we assume no SPAM error, and in practice similar results can be achieved by applying other SPAM-insensitive process tomography approaches to the CR-CNOT[47–49].

We compare the PAPA reconstruction for a noisy CR-CNOT to the actual simulated noisy CR-CNOT by calculating the trace distance between the Choi state of the three-qubit PAPA reconstruction, $\rho_{\mathcal{E}}$, and that of the actual process, $\rho_{\mathcal{E}}^{\text{act}}$. The results of our simulations and reconstructions (both done in MATLAB[55]) are shown in Fig. 4. As can be seen, for all values of the coherent error tested the PAPA reconstruction is approximately an order of magnitude closer to the noisy gate than the ideal gate (used as the initial guess). Further numerical simulation tests can be found in Supplementary Note 5.

**Continuous time evolution.** PAPA is intended to characterize short depth circuits made from discrete gates, following the circuit model of quantum computation, or digital quantum simulation. However, a natural question is how it could be used to characterize continuous evolution of a two-local Hamiltonian (or more generally a Lindbladian) with unknown parameters. While such a model contains the same number of free parameters as the PAPA ansatz, we do not believe they describe equivalent subsets of completely positive trace-preserving (CPTP) maps.

However, PAPA can be used to characterize short-time evolution of the two-local Hamiltonian, as the first order

Zassenhaus formula approximation to such evolution

$$e^{-it\hat{H}_{2-\text{local}}} \approx \prod_{j,k}^{N-1,N-j} \exp\{-it\hat{H}_j \otimes \hat{H}_{j+k}\}, \qquad (11)$$

is exactly of PAPA form (each element of the product on the right-hand side is a two-body process $\mathcal{E}_{j,j+k}$). Then, using the Lie-Trotter product formula, evolution for an arbitrary time can be simulated using the PAPA characterized short-time process.

## Discussion

Our physically motivated pairwise perturbative ansatz is an efficient and SPAM-insensitive approach to quantum process tomography that relies on fitting tomographic data to a constrained ansatz for the unknown quantum process. It requires only two-qubit process tomography on all pairs of qubits, such that the total number of free parameters scales only quadratically with qubit number. Further, our ansatz inherits SPAM-insensitivity from SPAM-insensitive two-qubit tomography, such as gate set tomography[39] or RB gate tomography[47–49].

The experimental demonstration of PAPA shows a significant improvement in the accuracy of reconstructed two-qubit processes calculated from the bootstrapped three-qubit process. Testing via numerical simulations validates the usefulness of our tomographic approach on the experimentally relevant CR-CNOT. In typical cases, the resulting description of the unknown quantum process found by our ansatz is an order of magnitude more accurate than the naïve initial guess. The accuracy of the PAPA reconstructions is set by the specifics of the classical numerical algorithm implemented (see Supplementary Note 6). In the future, we hope to improve the efficiency and accuracy of the classical algorithm underlying the PAPA reconstruction method[56,57].

It is worth noting that while we have chosen to build our ansatz for an $N$-qubit process from two-qubit processes, similar ansatz can be created from $K$-qubit processes for any $K < N$. These have experimental resource requirements that scale as a polynomial of order $K$, and are therefore still asymptotically efficient. We focus on case $K = 2$ in this work as two-qubit process tomography is within current experimental capabilities. For larger system sizes, there will be an optimal $K > 2$ that reduces the number of $K$-qubit subsets, and maintains a small enough $K$ to be experimentally feasible.

Finally, we comment briefly on the situations where PAPA may fail. Numerical reasons aside, PAPA reconstruction fails when the process being estimated is an operation that is not factorable to 2-body, or when non-Markovian noise is present. As such, PAPA reconstruction can be used as a form of model testing for error processes that entangle >2 qubits, or non-Markovian noise sources such as slow parameter drift. Similarly, PAPA+GST puts greater restrictions on the gate and context independence of the noise sources, and can be used as a model testing procedure for these error sources. This highlights the usefulness of ansatz-based approaches to QPT: even when they fail they provide useful information about the system.

## Methods

### Characterizing the two-qubit processes.
In the most general version of QPT, there is a completely unknown quantum process, which one wishes to determine. Applying PAPA to this problem, the required two-qubit QPT is derived from the form of Eq. (6). For a pair of qubits defined by the subset $\mathcal{S}$ we perform two-qubit QPT to characterize the effective process the qubits in $\mathcal{S}$ experience when the unknown process $\mathcal{E}$ is implemented on all $N$ qubits (with all other qubits initialized in the maximally mixed state), as depicted in Fig. 1a).

To see that Eq. (6) describes a valid two-qubit process, we describe the unknown $N$-qubit process in a basis of $N$-qubit processes as

$$\mathcal{E} = \sum_i \epsilon_i \bigotimes_{k}^{N} \Lambda_{i_k}, \qquad (12)$$

where $\Sigma \epsilon_i = 1$. Substituting this expression into the partial trace in Eq. (6), we obtain (recall $\mathcal{S} = \{m, p\}$)

$$\begin{aligned}
&\text{Tr}_{/\mathcal{S}}\left[\mathcal{E}\left(|\psi_{\mu_\mathcal{S}}\rangle\langle\psi_{\nu_\mathcal{S}}| \otimes \mathbb{I}_{N-2}\right)\right] \\
&= 2^{N-2} \sum_i \epsilon_i \Lambda_{i_m} \otimes \Lambda_{i_p}\left(|\psi_{\mu_\mathcal{S}}\rangle\langle\psi_{\nu_\mathcal{S}}|\right)\text{Tr}\left[\bigotimes_{k}^{N} \Lambda_{i_k}\left(\frac{\mathbb{I}}{2}\right)\right] \\
&= 2^{N-2} \sum_i \epsilon_i \Lambda_{i_m} \otimes \Lambda_{i_p}\left(|\psi_{\mu_\mathcal{S}}\rangle\langle\psi_{\nu_\mathcal{S}}|\right) \\
&\equiv 2^{N-2} \Lambda_\mathcal{S}\left(|\psi_{\mu_\mathcal{S}}\rangle\langle\psi_{\nu_\mathcal{S}}|\right),
\end{aligned} \qquad (13)$$

where we have defined $\Lambda_\mathcal{S} = \sum_i \epsilon_i \Lambda_{i_m} \otimes \Lambda_{i_p}$. The reduced Choi state can then be written as

$$\rho_\mathcal{S} = \frac{1}{2^2}\sum_{\mu_\mathcal{S}\nu_\mathcal{S}} |\psi_{\mu_\mathcal{S}}\rangle\langle\psi_{\nu_\mathcal{S}}| \otimes \Lambda_\mathcal{S}\left(|\psi_{\mu_\mathcal{S}}\rangle\langle\psi_{\nu_\mathcal{S}}|\right), \qquad (14)$$

and it is clear that $\Lambda_\mathcal{S}$ must describe a valid quantum process. We have labeled the reduced two-qubit processes as $\Lambda_\mathcal{S}$ to distinguish them from the two-qubit processes that construct the PAPA, $\mathcal{E}_{k,n+k}$ in Eq. (2), as $\Lambda_\mathcal{S}$ depend on the free parameters from all $\mathcal{E}_{k,n+k}$, not just those in the qubit subset $\mathcal{S}$.

In Eq. (6) we see that the qubits outside the qubit pair of interest (the spectator qubits) must be prepared in the maximally mixed state. If this is experimentally challenging, one can instead randomly sample spectator qubit preparations from the uniform distribution of the set of spectator qubit logical states. With sufficient sampling to generate accurate statistics, the normalized sum of the randomly sampled preparation states approaches the maximally mixed state for the spectator qubits. Thus, performing two-qubit QPT on the qubit pair of interest with spectator qubits prepared in a random logical state will characterize the desired effective process in Eq. (6).

### Experimental procedure.
Gatestrings are generated with pyGSTi[50], transpiled into our QGL[58] language and finally compiled to a hardware specific format for our custom APS2 arbitrary waveform generators[59]. To insure each GST experiment (across all three pairs) is subject to the same environmental noise on average, and as consistent as possible with other experiments, gatestrings from the three sets are interleaved on a shot-by-shot basis before being executed. This prevents long term drift from changing system conditions across gatestrings, or pairs of qubits. Additionally, to mimic the preparation of the spectator in the maximally mixed state (see Fig. 1a)), each two-qubit GST experiment is repeated an equal number of times with the third qubit starting in either $|0\rangle$ or $|1\rangle$. The results are then combined and analyzed irrespective of the state of the spectator.

Experimental data is passed back to pyGSTi for reconstruction. Details of the reconstruction process can be found in refs. [39] and [50]. To ensure viability of the PAPA process, the CPTP constraint is enforced at every iteration $L$ as new data is added. This guarantees a physical and consistent gate set is reconstructed. The downside with this requirement is a considerable increase in runtime and RAM necessary for GST to converge. To make this process tractable and time efficient, we used a Google Cloud instance[60] with 40 vCPUs and 961 GB of RAM, which allows all three GST data sets to be analyzed simultaneously. The upper bound of 961 GB is not tight and was chosen out of an abundance of caution to ensure convergence.

The 27 three-qubit gates were reconstructed using the Julia implementation of PAPA on a 24 vCPU workstation with 32 GB of RAM. The reconstruction took an average of 8 h for each three-qubit gate, using three cores and on the order of a GB of RAM. The reconstructions were run in parallel to reduce overall runtime.

Qubit control and readout are performed through dedicated coplanar waveguide (CPW) resonators, one coupled to each qubit[9]. $Q_1$ is equipped with a Josephson parametric amplifier from UC Berkeley[61] and $Q_3$ with a Josephson parametric converter from IBM[62] for improved readout fidelity. All the pulses are generated using the BBN pulse generators introduced in ref. [59]. Readout signals are acquired and processed using two Innovative Integration X6-1000 digitizers programmed with the BBN QDSP firmware (ibid). In particular, for each measured qubit, a 2.3 µs homodyne signal is integrated using a pre-calibrated matched filter[63], and subsequently reduced to a single-bit value according to an optimized threshold. All of this signal processing takes place on the digitizer field-programmable gate array (FPGA) board, thus expediting data acquisition and writing to disk. The results, which are digitized independently for each qubit, are then converted into a number of counts for each of the four combinations in a qubit pair, which is the input format for pyGSTi. This process is illustrated in Fig. 5.

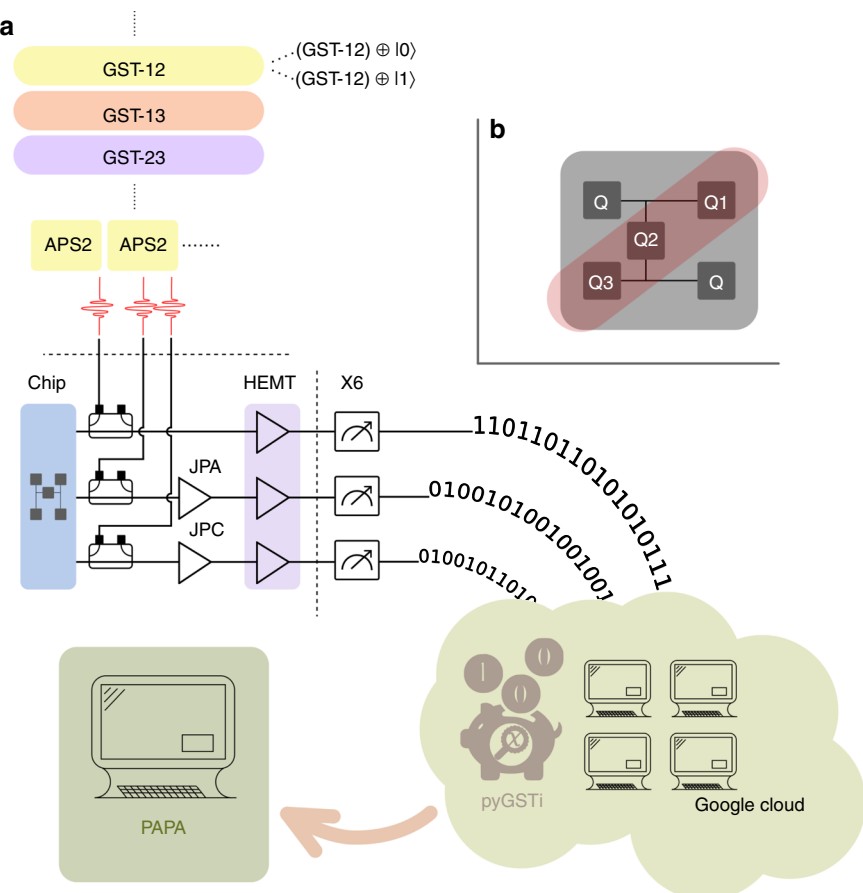

**Fig. 5 PAPA experimental data flow. a** GST experiments are interleaved by pair and spectator qubit state and sent to the BBN Arbitrary Pulse Sequencers (APS2)[59]. Gate instructions are converted to control pulses and sent to the five-qubit device. Demodulated readout signals are first amplified by HEMTs, JPAs and JPCs then digitized into qubit state information with custom firmware running on an X6-1000M digitizer card. This data is then passed to pyGSTi and reconstructed in parallel using Google Cloud Compute services. The three separate reconstructions are then passed to the PAPA algorithm for bootstrapping to three-qubit processes. **b** A notional diagram of the IBM five-qubit device used in the experiment. The location of the {Q1, Q2, Q3} subset is specified in red. Lines denote static capacitive coupling through CPW resonators.

**Gate set gauges in PAPA+GST.** Gauge freedom in GST gate set characterizations arises from the fact that the gate set, preparation, and measurement are characterized simultaneously using data that takes the form of expectation values such as

$$\text{data} = \langle\langle M|\hat{\mathcal{G}}|\rho\rangle\rangle, \tag{15}$$

where we have used super-operator notation for the preparation state $\rho$ and measurement observable $\hat{M}$. Such expectation values are invariant under transformations of the form

$$\text{data} = \langle\langle M|\hat{T}\hat{T}^{-1}\hat{\mathcal{G}}\hat{T}\hat{T}^{-1}|\rho\rangle\rangle, \tag{16}$$

where $\hat{T}$ is an invertible matrix referred to as the gauge matrix. This emphasizes that while gauge has no effect on physical observables and experiments, it can change the properties of a gate set (and change the preparation and measurement to compensate).

GST minimizes the effects of gauge freedom by gauge optimizing the characterized gate set to the a priori assumed ideal gate set, and it is these gauge-optimized gate sets that we have used for PAPA reconstruction in the Results section. We have also tried PAPA reconstruction with the GST characterizations that have not been gauge optimized to the ideal gate set, and as Fig. 6 shows, this results in a lower quality PAPA reconstruction.

As trace distance is a gauge variant quantity, the large trace distances for non-gauge-optimized GST data is a non-physical error, coming from the input data to the PAPA algorithm being gauge inconsistent across qubit pairs. To explore the limits of performance, we gauge optimize the PAPA reconstructions directly to the GST characterized gate sets before calculating the trace distance. The gauge optimization was done using pyGSTi, and the result plotted in Fig. 6 shows only modest improvements at best. We have also had mixed success taking the opposite approach: gauge optimizing the GST characterized pairwise gate sets to be in a consistent single-qubit gauge before performing the PAPA reconstructions, which we detail in Supplementary Note 4.

On average, we find the best approach to gauge consistency is to gauge optimize the GST reconstructions to the ideal gate set (as is done in pyGSTi by default), and use this for the PAPA reconstruction, which is what we present in the Results section. Future developments in using PAPA or other bootstrapping approaches on QPT data that has gauge freedom (such as from GST) will seek to address the gauge consistency issue by performing some parts of the initial pairwise tomographic reconstruction jointly.

**Simulation test error models.** For the results shown in Fig. 4, the unitary describing the CR-CNOT in the presence of coherent error is given by $\hat{U}_{\text{CNOT}} = \hat{Z}^1_{-90}\hat{X}^2_{90}\hat{U}_{\text{CR}}$, with the ±90° single-qubit rotations assumed to be perfect, and

$$\hat{U}_{\text{CR}} = \exp\left(-i\left[\left(\frac{\pi}{2}+\beta\right)\frac{\hat{Z}\hat{X}\hat{\mathbb{1}}}{2} + \phi\frac{\hat{\mathbb{1}}\hat{Z}\hat{Z}}{2}\right]\right), \tag{17}$$

where for compactness of notation we have suppressed the tensor product symbols, such that $\hat{Z}\hat{X}\hat{\mathbb{1}} = \hat{Z}\otimes\hat{X}\otimes\hat{\mathbb{1}}$.

In Eq. (17), the angles $\beta$ and $\phi$ quantify the coherent error, with $\beta$ the angle of over-rotation from the desired CR interaction between qubits 1–2, and $\phi$ the angle quantifying the effect of spurious $ZZ$-coupling between qubits 2–3. We consider the echoed CR-pulse of ref. [64], such that the only remaining $ZZ$-coupling is between the target and idle qubits (i.e., 2 and 3). We use values of $\beta$ between $\pi/16$ and $\pi/8$ radians, which produce non-ideal gates with trace-overlap fidelity of 95–99%, and values of $\phi$ between $10^{-3}$ and $4\times10^{-3}$ radians. For a gate of 400 ns in duration, these values of $\phi$ correspond to spurious $ZZ$-couplings of 2.5–10 kHz.

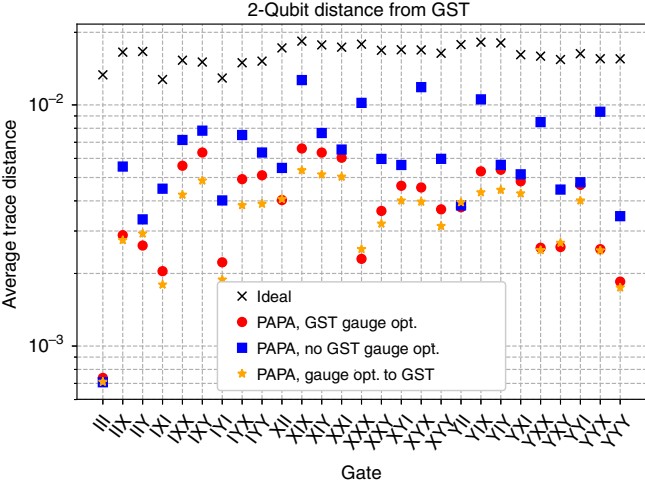

**Fig. 6 Comparison of the GST measured process matrices to the ideal process matrices, and to a variety of PAPA reconstructions of the experimental data.** Red circles are the data points from Fig. 3 with GST gauge optimization, and blue squares have no GST gauge optimization. Orange stars have a further round of gauge optimization performed between the PAPA reconstructions and the GST measured process matrices. As in Fig. 3, PAPA data points are the average trace distance of the three reduced processes from there corresponding GST characterizations.

## Data availability

All data presented is available on request.

## Code availability

Source code for the pairwise perturbative ansatz bootstrapping technique is available at https://github.com/BBN-Q/PAPA.jl.

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

## Acknowledgements

We acknowledge useful discussions with David Poulin. The qubit device and amplifiers used in the experiment were graciously provided by IBM under the Intelligence Advanced Research Projects Activity (IARPA) contract W911NF-16-0114. This material is based upon work supported by the U.S. Army Research Office under Contract No: W911NF-14-C-0048. Any opinions, findings and conclusions or recommendations expressed in this material are those of the authors and do not necessarily reflect the views of the U.S. Army Research Office.

## Author contributions

L.G. and H.K. developed the ansatz, and the characterization methodology. L.G. developed the bootstrapping algorithms used for the experimental data and numerical simulations. G.R., D.R., and M.W. performed the experiments, and processed the experimental data. All authors contributed to the writing of the manuscript.

## Competing interests

The authors declare no competing interests.
