## [Peer Review File · Nature Communications]

Reviewers' Comments:

Reviewer #1:

Remarks to the Author:

The manuscript discusses a new method for a simplified quantum process tomography, in which the full process tomography is replaced by process tomography for all pairs of qubits. In my opinion, this procedure is not physically justified (in contrast to authors' opinion), and therefore its results are not supposed to be well-correlated to the results of actual process tomography. In the paper the method is applied only to 3-qubit processes and, as I understood the result, this method is only shown to be better than not doing any tomography at all. The method does not give protection from the State Preparation And Measurement errors (SPAM), and therefore in order to be SPAM-insensitive, it has to rely on other methods. The experiment is not impressive (the qubit device was supplied by another group, parametric amplifiers for 2 qubits were supplied by other groups, while the third qubit even did not have a good amplifier and a HEMT was used instead).

Concluding, I do not see sufficient justification for the manuscript to be publishable in Nature Communications. However, I would recommend publishing this manuscript, for example, as a regular article in PRA.

Reviewer #2:

Remarks to the Author:

Please see attached file for comments.

Reviewer #3:

Remarks to the Author:

This paper develops a method for quantum process tomography called PAPA, where an N-qubit process is approximated by a sequence of 2-qubit processes.

Strengths of the paper:

The paper is motivated by an important practical need: how to characterize errors in N-qubit quantum information processors in a way that efficient enough to handle large N?

The PAPA method described here is very natural, since it is physically reasonable to assume that the dynamics of N qubits will not contain arbitrary many-particle correlations, but will be built out of combinations of few-particle (e.g., pairwise) correlations.

The experimental demonstration using superconducting qubits is quite impressive, as it deals with a number of technical difficulties (such as using GST to compensate for the effects of SPAM errors), and it shows that the PAPA method can describe the noise in this system with noticeably better accuracy than naive models.

Weaknesses of the paper:

While the general idea that underlies PAPA is very natural, there are several specific details of the PAPA method that need more explanation and motivation, to justify why this is the "right" approach.

For instance, it is clear that we want to describe an N-qubit process in terms of dynamics on pairs of qubits. Physically, the most natural way would be to use a 2-local Hamiltonian, or (for Markovian noise) a Lindblad operator with 2-local terms. However, PAPA does something different:

it uses a sequential composition of 2-qubit CPTP maps. This raises several questions: since these maps do not commute with each other, what is the right order to apply them? Do we need to apply each one multiple times, like a Lie-Trotter expansion? In many cases (such as describing a noisy gate in a quantum information processor) most of these maps will be close to the identity; in that case, the maps "almost" commute, so perhaps this makes the ordering less important?

At a high level, the authors are trying to argue that the PAPA approach is both highly expressive (in that it can model physically realistic noise processes) and computationally efficient (in that it requires only polynomial resources as a function of the number of qubits N). However, the authors don't seem to complete some important parts of this argument. For instance, how accurately can the PAPA ansatz (equation 2) describe time-evolution with a 2-local Hamiltonian? And given an N -qubit channel in "PAPA form," how efficiently can one compute the reduced Choi states (equation 3), in order to compare them with experimental data?

Finally, I think some of the details in the "supplementary information" (about the numerical optimization used to fit the PAPA model to the data) may be incorrect. First, the paper says that equation (11) encodes the CP (complete positivity) constraint, but I believe this is actually the TP (trace preservation) constraint? More importantly, the paper says that equation (14) is a semidefinite program, but I could not see why this is true? It seems to me that the reduced Choi states of the N -qubit quantum channel specified by PAPA are high-degree polynomials of the PAPA parameters, and in general these are not convex. Is there some additional approximation being used here, which replaces these non-convex functions with functions that are convex?

Overall evaluation:

I think this paper should not be accepted as is, but it could be accepted after some revisions. I believe the basic idea behind the paper is sound, but the theory sections of the paper could be improved.

Response to referee comments for “Bootstrapping quantum process tomography via a perturbative ansatz”

L. C. G. Govia,^{*} G. J. Ribeill, D. Ristè, M. Ware, and H. Krovi
Raytheon BBN Technologies, 10 Moulton St., Cambridge, MA 02138, USA

Our responses to the referees follow, and a list of changes can be found at the end of this document.

Sincerely,

Luke Govia, Guilhem Ribeill, Diego Ristè, Matthew Ware, and Hari Krovi

I. REFEREE 1

It is unfortunate that referee 1 took a negative view of our manuscript. We believe their concerns are largely due to misunderstandings of our work. In the following, we address each of their concerns in detail (referee’s report in italics, our responses in standard font).

The manuscript discusses a new method for a simplified quantum process tomography, in which the full process tomography is replaced by process tomography for all pairs of qubits. In my opinion, this procedure is not physically justified (in contrast to authors’ opinion), and therefore its results are not supposed to be well-correlated to the results of actual process tomography.

While we respect the referee’s opinion, we are unable to concretely assess or respond to it as the referee has not provided any reasoning or justification for their opinion. More importantly, the results of our work show that our choice of ansatz is justifiable, given the accuracy we obtain in characterization. Further, neither referee 2 nor 3 had any concerns with the physical justification of our ansatz, and referee 3 went so far as to call this out as a *strength* of our work, when they say, “The PAPA method described here is very natural, since it is physically reasonable to assume that the dynamics of N qubits will not contain arbitrary many-particle correlations, but will be built out of combinations of few-particle (e.g., pairwise) correlations.”

While no ansatz can ever be all encompassing of physical phenomenon, we believe that we have sufficiently justified our choice of ansatz in the manuscript. This is supported by both our results, and the opinions of referees 2 and 3. As such, we believe that no change in the manuscript is necessary.

In the paper the method is applied only to 3-qubit processes and, as I understood the result, this method is only shown to be better than not doing any tomography at all.

We believe that our results were misunderstood. The fact that this method provides reconstruction of three-qubit processes is the main result. We cannot compare to other methods for three-qubit tomography because without an ansatz, it is not possible to do any kind of three-qubit process tomography. More importantly, we compare the PAPA reconstructions to the GST pairwise characterizations of the experimental data, and show that they give better agreement than the ideal gates (“doing no tomography at all”). Thus, we directly compare our reconstructions to experimental data. Further, in our numerical simulations we are able to directly compare to the exact process, and demonstrate the accuracy of our ansatz.

Three-qubit process reconstruction is the natural first demonstration of the PAPA approach. Even at three qubits this is already beyond the scope of naïve process tomography, and there is nothing preventing eventually applying the PAPA approach to more than three qubits, as we have already done for experimental collaborators (not discussed in this manuscript).

The method does not give protection from the State Preparation And Measurement errors (SPAM), and therefore in order to be SPAM-insensitive, it has to rely on other methods.

^{*} luke.c.govia@raytheon.com

The referee is indeed correct that PAPA does not have any intrinsic SPAM-insensitivity, but it is not meant to. It is a method for bootstrapping small-scale tomographic data to large scale reconstructions of multi-qubit quantum processes. The fact that it is compatible with SPAM-insensitive two-qubit tomography methods is sufficient, it does not need to be SPAM-insensitive itself.

The experiment is not impressive (the qubit device was supplied by another group, parametric amplifiers for 2 qubits were supplied by other groups, while the third qubit even did not have a good amplifier and a HEMT was used instead).

Concluding, I do not see sufficient justification for the manuscript to be publishable in Nature Communications. However, I would recommend publishing this manuscript, for example, as a regular article in PRA.

We suspect that part of the referee’s concern is simply a field-specific cultural misunderstanding. It is commonplace in the circuit QED community to use amplifiers and qubit devices fabricated by other groups. In fact, the qubit device used was part of on-going collaborations with the group that fabricated it.

However, we are confused as to why the referee would judge an experiment “not impressive” simply because we did not fabricate the devices used in the experiment. There is a lot of technical difficulty that goes into any circuit QED experiment, far beyond the fabrication. With no offense intended to the hard work required to fabricate devices (which we are well aware of, given our in-house fabrication), the device fabrication is likely the *least* impressive part of this experiment.

For instance, the control stack required to interleave multiple two-qubit GST experiments is highly nontrivial and sophisticated. Our ability to perform these experiments is a results of years of internal research, design, and production of high-speed control electronics, with their accompanying firmware and software. Details of this work can be found in Ref. [59] of the main text. It is worth pointing out that this is one of the first works in the literature to demonstrate two-qubit GST.

Further, referee 3 again has the opposite opinion of referee 1, and appreciates the technical effort required for GST and our experiment. They explicitly state, “The experimental demonstration using superconducting qubits is quite impressive, as it deals with a number of technical difficulties (such as using GST to compensate for the effects of SPAM errors), and it shows that the PAPA method can describe the noise in this system with noticeably better accuracy than naive models.”

Moreover, from a high-level perspective this experiment is not meant to be hard, otherwise the PAPA approach would have less broad applicability. The fact that we could implement the experiment directly with our existing control stack is a beneficial feature of our work.

II. REFEREE 2

We thank the referee for their strong endorsement of our manuscript, and their very detailed and helpful report. The referee raises several important considerations both broadly for PAPA, and for the use of GST collected data with the PAPA bootstrapping technique. In what follows, we attach their complete report, and then address each of their questions individually. We indicate the corresponding changes made to the manuscript in our responses.

Report of Referee 2

In their work "Bootstrapping quantum process tomography via a perturbative ansatz" the authors tackle a critical question for scaling up many-qubit characterization techniques, namely, how to overcome the curse of dimensionality. A polynomial-scaling tomographic protocol is of serious import and interest; this work well-motivated, important, and well-presented. I think it is probably suitable for publication in Nature Communications. However, there are a few issues that should be addressed first prior to publication.

A. Issues relating to gauge

The authors have chosen to use gate set tomography (GST) as the tomographic engine feeding their PAPA protocol. Given various "nice" properties of GST, I think this choice makes a lot of sense. However, using GST may cause a non-trivial problem with the PAPA reconstruction. One advantage of GST over other tomographic protocols is that it does not assume perfectly calibrated input states and measurements. Rather it self-consistently characterizes all gates, state preparations, and measurements in a gate set simultaneously. However, the cost of this self-consistency is the gauge degree of freedom: Given a gate set with gates G_i , states $|\rho_j\rangle\rangle$ and measurements $\langle\langle E_k|$, the probability of observing outcome E_k , conditioned on input ρ_j and gate sequence S of $G_{S_1} \dots G_{S_m}$, is given by the weighted inner product $\langle\langle E_k | G_{S_m} \dots G_{S_1} | \rho_j \rangle\rangle$. However, given any invertible matrix T , we may construct a new gate set with gates TG_iT^{-1} , states $T|\rho_j\rangle\rangle$, and measurement outcomes $\langle\langle E_k|T^{-1}$. By inspection, we see that these two gate sets will yield the same probability predictions for gate sequence S , and indeed for *any* gate sequence, even though the constituent matrix and vector descriptions of the gates, measurements, and states are manifestly different in the two cases.

Gauge degrees of freedom are not a problem if one only wishes to take one gate set and use it to predict outcome probabilities; by definition, the gauge in such cases is necessarily undetectable. Problems arise when either a) one wishes to report gauge-variant quantities (e.g., gate fidelities, diamond distances, etc.), or b) one wishes to "stitch together" different gate sets. The former issue is well-addressed through gauge optimization, in which, using only gauge transformations, one attempts to make a gate set appear as close as possible to another gate set. This functionality is discussed in the literature and included in the pyGSTi software package used by the authors.

The latter issue is more of an open question, as far as I know. If we restrict ourselves to even just a single-qubit case, we can see that there can be problems. For example, suppose I have a black box with three buttons on it, labeled A, B, and C, respectively; each button implements a single-qubit gate operation. Suppose I perform GST on {A, B}, and I conclude that $A=X_{\pi/2}$, and $B=Z_{\pi/2}$. I now perform GST on {B, C} and I conclude that $B=Z_{\pi/2}$ and $C=X_{\pi/2}$. I could easily conclude then that A and C implement the same operation. However, there is another equally valid explanation—because $\{X_{\pi/2}, Z_{\pi/2}\}$ is gauge-equivalent (by conjugation of $Z_{\pi/2}$) to $\{Y_{\pi/2}, Z_{\pi/2}\}$, it is also possible that A, B, C are all unique $\pi/2$ rotations about three mutually

orthogonal axes (e.g., $X_{\pi/2}$, $Y_{\pi/2}$, $Z_{\pi/2}$). In order to actually determine how all three buttons behave it is necessary to characterize them all simultaneously. (Note also that, because the gauge transformation is given by $T=Z_{\pi/2}$, if state preparation and measurement are in the computational basis (i.e., the Z basis), consistent descriptions of the state preparation and measurement operations will be given in all cases, so while inconsistent SPAM descriptions can be a signature of gauge inconsistencies, there can also be gauge inconsistencies that do not modify SPAM.)

The challenge in this manuscript is admittedly a bit different than and not as problematic as the above example. However, I think one can still run into trouble. For example, suppose that a three-qubit process matrix is, in some gauge, $X \otimes X \otimes X$. (Here I am following standard convention and the authors' - "X" refers to X_{π} .) GST is done on all three pairs of cubits. If, for all three pairs, $X \otimes X$ is reported, then there is no problem- following the least-squares minimization prescribed by Eq. 18, $X \otimes X \otimes X$ should be recovered.

However, suppose that instead, GST reports for $\{A, B\}$ and $\{A, C\}$ $X \otimes X$, but for $\{B, C\}$ $Y \otimes Y$. This is a perfectly valid choice for GST to make, as $X \otimes X$ is gauge-equivalent to $Y \otimes Y$ (via local conjugation by $Z_{\pi/2}$). However, I strongly suspect that, were such GST results to be fed into PAPA, the resulting three-qubit process matrix probably would *not* be gauge-equivalent to $X \otimes X \otimes X$. (This is because the search is being done over all parameters in the ansatz, (i.e., Eqs. 2 and 3) and the optimizer will be forced to pick parameters that correspond to one-qubit process matrices that are "in between" X and Y. Such operators in general will not be gauge-equivalent to them, and possibly not even unitary.)

I will admit that, due to some property of the ansatz space, it is *possible* that gauge-equivalence is preserved when gauge-transformed inputs are fed into PAPA. However, if this is the case the authors should demonstrate this. (Ideally this would be done analytically, but alternatively, numerical demonstration on a variety of different cases would also offer compelling evidence.)

However, my strong intuition is that this is *not* the case, that is, (to abuse notation slightly) $\text{PAPA}[XX, XX, XX]$ is not gauge-equivalent to $\text{PAPA}[XX, XX, YY]$; these three-qubit process matrices would predict different outcome probabilities for the same circuit.

If this is the case, then some small modification to PAPA is probably in order. I am not certain what the simplest solution is, but one option would be to transform the GST outputs into a *gauge-consistent* collection of gate sets before they are processed by the PAPA optimization routine. It is also possible that folding gauge into the search space of Eq. 18 could help, but it is not immediately obvious to me how that would work.

To conclude my comment regarding gauge, what I believe is necessary for PAPA to be gauge-resilient is a demonstration that, independent of whatever gauges the inputs to PAPA are given in, the PAPA output predicts the same outcome probabilities for

arbitrary multi-qubit gate sequences.

B. Metrics and performance of PAPA under ideal circumstances

I am also a little confused by the demonstrations of PAPA's viability in the paper.

1. While I think trace distance is a good way to show (e.g., in Fig. 3) that PAPA doesn't distort the GST fits too badly, it doesn't demonstrate that the three-qubit PAPA reconstruction is actually trustworthy. In other words, how well does the PAPA reconstruction actually fit the data? This can be tackled in a variety of ways, but they all essentially boil down to computing log-likelihood ratios on the data to estimate a goodness of fit via a χ^2 test. (This can be implemented by hand, or, alternatively, the pyGSTi package offers a method for implementing this test directly; see in the pyGSTi package directory `jupyter_notebooks/Tutorials/algorithms/ModelTesting.ipynb`.)

(Additionally, I'll note that because the experimental data shows some non-Markovian behavior (as shown in Fig. 2), I think a goodness-of-fit test on simulated data would be helpful as well, so we could see how we should expect PAPA's goodness-of-fit to behave when there's no underlying non-Markovianity.)

2. While I think the numerical simulations do a nice job demonstrating PAPA's viability, one thing I am left wondering is where the errors in Fig. 4 come from. How much is due to finite sample error and/or errors in the GST reconstruction? How much is due to PAPA not being able to find the global minimum? Should we expect, if we know the true two-qubit processes, that PAPA can reconstruct the three-qubit process matrix exactly? (I also think it is possible that some of this residual error is due to the aforementioned gauge issue, but I am not sure.) One way to help explain the answers would be to show how well PAPA performs (both by trace distance and goodness-of-fit) when it is given true two-qubit process matrices (removing any errors introduced by finite sampling or failure of GST to find the global optimum).

C. Minutiae

1. I would put the numerical demonstration before the experimental demonstration. This way the reader has a better sense as to what they should expect from experimental implementation.

2. While Figs. 4 and 5 explicitly state that they are from simulated data, Figs. 2 and 3 don't state they're from experimental data; I would explicitly call this out in the captions for Figs. 2 and 3.

3. I would move Eq. 18 out of the Methods section and present it relatively early in the main text, once the appropriate quantities are defined. This is the objective function that is at the heart of PAPA, so the reader should be aware of this as soon as possible. In general, presenting something like a block of pseudocode providing an end-to-end description of PAPA (either in the main text, methods, or supplemental material) would

be good, making it easier for a reader to reconstruct the work if she or he so desires.

Response to Referee 2

- *A. Issues relating to gauge*

To start, we would like to highlight that gauge issues are not an inherent aspect of PAPA, but of the use of tomographic data that has gauge freedom in a bootstrapping reconstruction technique such as PAPA. Thus, these gauge issues relate only to PAPA+GST. Of course, given the positive attributes of using GST as the underlying two-qubit tomography method, it is important to address the question of gauge in the PAPA reconstruction. We thank the referee for highlighting this fact, and the very detailed explanation in their report, which has made it much easier to address their concerns.

We have performed three new PAPA reconstructions (the results of which are all contained in the manuscript), and added a considerable amount of new text in regards to the gauge issue. In the following, we will briefly summarize our conclusions. As correctly pointed out by the referee, the main issue arises when the pairwise two-qubit gate sets characterized independently using GST are not gauge-consistent. In this case, the same physical gate, for example $\hat{X}_1 \otimes \mathbb{I}_2$ from gate set 1–2 and $\hat{X}_1 \otimes \mathbb{I}_3$ from gate set 1–3, may have different process matrix descriptions due to the inherent gauge freedom in GST characterizations.

The major result of our new reconstructions is that the gauge optimization routine intrinsic to `pyGSTi`, which seeks to gauge-optimize a gate set to the ideal, is sufficient to ensure approximate gauge-consistency across the pairwise gate sets. This is elucidated in Fig. 6 of the main text, where it can be seen that without this intrinsic gauge optimization the PAPA reconstructions are less accurate.

Further, we were unable to improve consistently upon the reconstructions given by the gauge-optimized to ideal gate sets from GST. As also shown in Fig. 6, even gauge-optimizing the reduced two-qubit gate sets from the PAPA three-qubit reconstructions directly to the GST gate sets does not improve the results consistently, or by a significant amount when it does. This is an example of post-processing the PAPA reconstructions for better gauge agreement, but we have found it to be mildly successful at best.

Conversely, we can pre-process the GST characterizations before performing the PAPA reconstruction, as we demonstrate in Section IV and Fig. 1 of the supplementary material. Here, we perform a gauge-optimization that ensures that all shared single-qubit gates between gate sets are as close to identical as possible (and then transform the entire gate set by this gauge transformation). We found that this improves the results more than the post-processing did (compare Fig. 6 of the main text to Fig. 1 of the supplementary material), but only for three-qubit gates with low weight. For gates with high weight this single-qubit gauge-optimization actually made the results worse.

- *B. Metrics and performance of PAPA under ideal circumstances*

1. In the updated manuscript Fig. 2, we now plot the N_σ and goodness-of-fit as calculated by `pyGSTi` for both the direct GST reconstruction, and the reduced two-qubit gate sets calculated from the PAPA reconstructed three-qubit gates (with the identity gate on the qubit not involved in the gate set). As can be seen, model violation increases, but to what we judge to be an acceptable level. We thank the referee for pointing out a useful additional metric to quantify the accuracy of the PAPA reconstructions.
2. In the numerical simulations of coherent error, or decoherence on ideal gates (originally contained in the main text but now moved to the supplementary material), the exact processes are of PAPA form, so an optimal algorithm should be able to reconstruct the exact process perfectly. There is no finite sampling error, or GST reconstruction for the simulations (only the experimental data), so any residual inaccuracy in the reconstruction is due to the PAPA numerical optimizer not being able to find the global minimum. Thus, the data shown in Fig. 2 of the supplementary material (moved from the main text) as well as Fig. 4 of the supplementary material, give a good indication of the numerical accuracy of the `MATLAB` implementation of PAPA, and we believe that the residual inaccuracy is entirely due to imperfect numerical optimization. The `Julia` implementation has shown improved numerical accuracy for the experimental data (see Fig. 1 of this response letter), but we have not tried it for the simulated data (as we did not believe this added sufficient value to the manuscript). Also, if asked to reconstruct an ideal gate, and seeded the ideal gate as the initial guess, then the PAPA algorithm is able to perfectly reconstruct this gate; as expected, since there should be no optimization *at all* in this case. However, if given a different initial guess, the accuracy of reconstruction is not perfect (by which we mean it does not reach machine precision), again highlighting inaccuracy from the numerical solver.

There is certainly more work to be done in improving the accuracy of the numerical solver used in PAPA. However, we believe this work to be beyond the scope of this manuscript, which introduces and demonstrates the first application of the PAPA method, and more suitable for follow-on work optimizing performance.

Figure 1. Comparison of the PAPA reconstruction accuracy (as measured by the trace distance to the GST data) for the MATLAB and Julia implementations of the codebase.

- *C. Minutiae*

1. We appreciate this suggestion from the referee, and understand their reasoning behind it. However, to accommodate additional theoretical analysis suggested by referee 3 we have shortened the numerical demonstration in the main text considerably, moving roughly half of it to the supplementary material. As such, we feel the current order, with experimental results first, remains the more appropriate one.
2. This change has been made, we thank the referee for pointing this out.
3. We thank the referee for this suggestion, as we feel it helps the flow of the manuscript. We have moved the objective function definition near the end of the section "Pairwise Perturbative Ansatz", and added pseudocode to the supplementary material.

III. REFEREE 3

We thank the referee for their comprehensive evaluation of our manuscript, and their useful comments and suggestions. It is encouraging to see that they appreciated the core motivation for PAPA, and the experimental effort that goes into even a process tomography approach designed to reduce this effort exponentially. Their comments and questions were very enlightening and useful, and in addressing their concerns the manuscript has been strengthened considerably. We now respond to each of their questions in detail (referee's report in italics, our responses in standard font), and indicate where the corresponding changes have been made in the manuscript.

This paper develops a method for quantum process tomography called PAPA, where an N -qubit process is approximated by a sequence of 2-qubit processes.

Strengths of the paper:

The paper is motivated by an important practical need: how to characterize errors in N -qubit quantum information processors in a way that efficient enough to handle large N ?

The PAPA method described here is very natural, since it is physically reasonable to assume that the dynamics of N qubits will not contain arbitrary many-particle correlations, but will be built out of combinations of few-particle (e.g., pairwise) correlations.

The experimental demonstration using superconducting qubits is quite impressive, as it deals with a number of technical difficulties (such as using GST to compensate for the effects of SPAM errors), and it shows that the PAPA method can describe the noise in this system with noticeably better accuracy than naive models.

Weaknesses of the paper:

While the general idea that underlies PAPA is very natural, there are several specific details of the PAPA method that need more explanation and motivation, to justify why this is the "right" approach.

For instance, it is clear that we want to describe an N -qubit process in terms of dynamics on pairs of qubits. Physically, the most natural way would be to use a 2-local Hamiltonian, or (for Markovian noise) a Lindblad operator

with 2-local terms. However, PAPA does something different: it uses a sequential composition of 2-qubit CPTP maps. This raises several questions: since these maps do not commute with each other, what is the right order to apply them?

For characterization of a shallow-depth circuit, the order of the PAPA ansatz should follow the order of two-qubit gates on each pair of qubits. We have added a note in the main text of this, and thank the referee for pointing out this oversight on our part. For characterization of short time evolution of a continuous two-local process (as now discussed in the manuscript) the order should not matter.

Do we need to apply each one multiple times, like a Lie-Trotter expansion? In many cases (such as describing a noisy gate in a quantum information processor) most of these maps will be close to the identity; in that case, the maps "almost" commute, so perhaps this makes the ordering less important?

In general, for shallow-depth circuits you would not have to construct a modified PAPA ansatz with repetition of pairwise processes. As detailed further in our next response, for simulation of continuous time evolution, you can use a Lie-Trotter expansion with a PAPA characterized short time evolution. The referee is correct that for maps close to the identity (such as weak-error maps) the ordering of the PAPA ansatz is less important.

At a high level, the authors are trying to argue that the PAPA approach is both highly expressive (in that it can model physically realistic noise processes) and computationally efficient (in that it requires only polynomial resources as a function of the number of qubits N). However, the authors don't seem to complete some important parts of this argument. For instance, how accurately can the PAPA ansatz (equation 2) describe time-evolution with a 2-local Hamiltonian?

As we envisioned it, PAPA was not meant to describe evolution of a two-local Hamiltonian up to arbitrary evolution times. It was meant to describe a piece-wise circuit of two-local gates applied one after another, and to capture unintended spurious two-body operations that occur during these gates. In practice, each of these gates is a two-local interaction for a given evolution time, but importantly, the *intended* two-local interactions for all qubits are not acting at the same time. This circuit picture of the quantum process is applicable to most current approaches to quantum computing, and to digital quantum simulation.

Returning to the situation the referee describes, that of the time-evolution of a 2-local Hamiltonian (or 2-local Lindbladian in general). If there is time-dependence in this setup, then PAPA is not an appropriate ansatz except at very short times, since a fully time-dependent 2-local Hamiltonian is universal. If there is no time-dependence, then the number of free parameters in such an evolution is exactly the same as the number of free parameters in the PAPA ansatz.

This fact might lead one to believe that this means a single PAPA ansatz can perfectly describe such 2-local evolution. However, we do not believe this is the case. Instead, as the referee suggests, using the Lie-Trotter expansion on the 2-local process, we obtain a sequence of operations that has PAPA form, repeated many times. One can use PAPA to characterize the generating element of this sequence, which can be done by characterizing the 2-local process for short evolution time. Using this PAPA-characterized process, one can then describe arbitrary time evolution via the Lie-Trotter product formula. We describe this in the updated manuscript.

And given an N -qubit channel in "PAPA form," how efficiently can one compute the reduced Choi states (equation 3), in order to compare them with experimental data?

This computation is efficient in the sense that it only involves a polynomial number of matrix multiplications (one for each component in the PAPA ansatz) and a partial trace operation. However, the naïve way of doing this is not asymptotically efficient as the number of qubits grows, as you must perform at least one matrix multiplication with a matrix describing the full system of N qubits, and the size of this matrix grows exponentially with the number of qubits.

The existing numerical implementations of PAPA are therefore currently not appropriate for characterizing systems of N qubits where N is so large that it is not possible to store a $2^N \times 2^N$ matrix in memory, or operate with such a matrix. However, all of this assumes a Schrödinger-style calculation of the operation

$$\mathcal{I} \otimes \mathcal{E} (|\Psi_{2N}\rangle\langle\Psi_{2N}|), \quad (1)$$

where $|\Psi_{2N}\rangle$ is the maximally entangled state on a system of $2N$ qubits. Feynman path integral approaches may allow a straightforward parallelization of this calculation, especially since the PAPA construction is naturally pairwise. We have added a short discussion of this in the Methods, and we hope to explore these ideas in future work.

Finally, I think some of the details in the "supplementary information" (about the numerical optimization used to fit the PAPA model to the data) may be incorrect. First, the paper says that equation (11) encodes the CP (complete positivity) constraint, but I believe this is actually the TP (trace preservation) constraint?

We thank the referee for catching this typographical error, which we have corrected.

More importantly, the paper says that equation (14) is a semidefinite program, but I could not see why this is true? It seems to me that the reduced Choi states of the N -qubit quantum channel specified by PAPA are high-degree polynomials of the PAPA parameters, and in general these are not convex. Is there some additional approximation being used here, which replaces these non-convex functions with functions that are convex?

In the previous version of the manuscript, we were using the definition of an SDP as an optimization problem over positive semi-definite matrices, which is true of our optimization problem over the χ -matrix parameters. We were aware that the reduced Choi states contain high-degree polynomials of these parameters, and therefore the search space is not obviously convex, as we alluded to with the statement below Eq. (14) (Eq.(21) in the updated version) "However, the operations involved in calculating C_1 are not obviously convex, and as a result the problem is not compatible with available convex-optimization packages. As such, we have not used this approach, but in future work hope to explore making the problem compatible with convex-optimization."

After additional literature search, it appears that the standard definition of an SDP is an optimization problem over positive semi-definite matrices with a convex cost function, as the referee correctly points out. This is not the case for the PAPA optimization, and as such we have removed the offending paragraph from the supplementary material.

Overall evaluation:

I think this paper should not be accepted as is, but it could be accepted after some revisions. I believe the basic idea behind the paper is sound, but the theory sections of the paper could be improved.

We thank the referee again for their detailed evaluation, and we believe we have improved the theory sections as they have suggested.

IV. LIST OF CHANGES

Major text additions and rearrangements are indicated in color in both the main text and supplementary material. We list here major text deletions and figure additions.

1. Figure 2 of the main text has been updated to include model violation plots for the PAPA reconstructions.
2. Figure 3 of the main text has been updated with the PAPA reconstructions generated from the Julia implementation of the codebase.
3. Figure 6 of the main text has been added, which shows a comparison of PAPA performance with and without two-qubit gauge optimization.
4. Part of the numerical simulation tests of PAPA have been moved from the main text to the supplementary material (the associated text is shown in color in the supplement).
5. Section II of the supplementary material has been newly added (to preserve formatting, it is not color-coded as the rest of the new text has been.)
6. Figure 1 of the supplementary material is new, and shows the performance of PAPA with and without one-qubit gauge optimization.
7. Figure 2 of the supplementary material has been moved there from the main text.
8. "The accuracy of the PAPA reconstructions is set by the specifics of the classical numerical algorithm implemented (see Supplementary Information). In the future, we hope to improve the efficiency and accuracy of the classical algorithm underlying the PAPA reconstruction method [54,55]." was moved from the end of the section "Simulation Tests of the Ansatz" to the second paragraph of the Discussion (replacing a similar set of sentences that was previously there).

9. “We note that this has the form of a semi-definite program (SDP). However, the operations involved in calculating C_1 are not obviously convex, and as a result the problem is not compatible with available convex-optimization packages. As such, we have not used this approach, but in future work hope to explore making the problem compatible with convex-optimization.” deleted from below what is now Eq. (21) of the supplementary material.

Reviewers' Comments:

Reviewer #2:

Remarks to the Author:

I thank the authors for their careful consideration of my comments, and the associated changes in their manuscript. I find their work now suitable for publication in Nature Communications.

Reviewer #3:

Remarks to the Author:

Review of "Bootstrapping quantum process tomography via a perturbative ansatz":

This is my second review of this paper, following revisions by the authors. I think these revisions have fully addressed my concerns about the paper.

In particular, I think the paper does a better job of explaining how PAPA can describe the time evolution that results from a quantum circuit or a local Hamiltonian. It seems clear that PAPA is most suitable for describing quantum circuits, rather than continuous-time evolution. The paper also clarifies several technical details of how one fits the PAPA model to the observed data.

Overall, I think the paper is now suitable for publication.

Dear Dr. Bentivegna,

Thank you for communicating the reviewer comments to us. Our response to their comments now follows.

Reviewer 2

“I thank the authors for their careful consideration of my comments, and the associated changes in their manuscript. I find their work now suitable for publication in Nature Communications.”

We thank the reviewer for their recommendation of publication, and are happy that they are satisfied with the changes made in the manuscript in light of their insightful first report.

Reviewer 3

“This is my second review of this paper, following revisions by the authors. I think these revisions have fully addressed my concerns about the paper.

In particular, I think the paper does a better job of explaining how PAPA can describe the time evolution that results from a quantum circuit or a local Hamiltonian. It seems clear that PAPA is most suitable for describing quantum circuits, rather than continuous-time evolution. The paper also clarifies several technical details of how one fits the PAPA model to the observed data.

Overall, I think the paper is now suitable for publication.”

We thank the reviewer for their recommendation of publication, and are glad to see that our revisions have fully addressed all their concerns about our manuscript.

Sincerely,

Luke Govia, Guilhem Ribeill, Diego Ristè, Matthew Ware, and Hari Krovi